# *p*-Coumaric Acid Differential Alters the Ion-Omics Profile of Chia Shoots under Salt Stress

**DOI:** 10.3390/plants13111564

**Published:** 2024-06-05

**Authors:** Mbukeni Nkomo, Mihlali Badiwe, Lee-Ann Niekerk, Arun Gokul, Marshal Keyster, Ashwil Klein

**Affiliations:** 1Plant Biotechnology Laboratory, Department of Agriculture, University of Zululand, Main Road, Kwadlagezwe 3886, South Africa; 2Plant Omics Laboratory, Department of Biotechnology, Life Science Building, University of the Western Cape, Robert Sobukwe Road, Bellville 7530, South Africa; mbadiwe@sun.ac.za; 3Environmental Biotechnology, Department of Biotechnology, Life Science Building, University of the Western Cape, Robert Sobukwe Road, Bellville 7530, South Africa; 3255882@myuwc.ac.za (L.-A.N.); mkeyster@uwc.ac.za (M.K.); 4Department of Plant Sciences, Qwaqwa Campus, University of the Free State, Phuthadithjaba 9866, South Africa; gokula@ufs.ac.za

**Keywords:** chia, salt stress, *p*-coumaric acid, mineral content

## Abstract

*p*-Coumaric acid (*p*-CA) is a phenolic compound that plays a crucial role in mediating multiple signaling pathways. It serves as a defense strategy against plant wounding and is also presumed to play a role in plant development and lignin biosynthesis. This study aimed to investigate the physiological and ionomic effect of *p*-CA on chia seedlings under salt stress. To this end, chia seedlings were supplemented with Nitrosol^®^ containing 100 μM of *p*-CA, 100 of mM NaCI, and their combined (100 mM NaCI + 100 μM *p*-CA) solutions in 2-day intervals for a period of 14 days along with a control containing Nitrosol^®^ only. The treatment of chia seedlings with 100 mM of NaCI decreased their growth parameters and the content of the majority of the essential macro-elements (K, P, Ca, and Mg), except for that of sodium (Na). The simultaneous application of *p*-CA and a salt stress treatment (*p*-CA + NaCI) alleviated the effect of salt stress on chia seedlings’ shoots, and this was indicated by the increase in chia biomass. Furthermore, this combined treatment significantly enhanced the levels of the essential microelements Mg and Ca. In summary, this brief report is built on the foundational work of our previous study, which demonstrated that *p*-CA promotes growth in chia seedlings via activation of O_2_^−^. In this brief report, we further show that *p*-CA not only promotes growth but also mitigates the effects of salt stress on chia seedlings. This mitigation effect may result from the presence of Mg and Ca, which are vital nutrients involved in regulating metabolic pathways, enzyme activity, and amino acid synthesis.

## 1. Introduction

Salt stress is primarily caused by improper agricultural practices, which have been estimated to affect 20% of all cultivated land and 33% of irrigated agricultural land worldwide [1]. Generally, plants grown in salt-stress-inducing environments are subject to major drawbacks, which may occur due to irregular irrigation, inadequate drainage, the application of the wrong fertilizer, and the use of sea water in mining [2,3]. In plants, the most common effects of salt stress include slow growth or inhibition, which mostly arise from the disturbance of nutritional macro elements [4,5]. In fact, long-term exposure to salt stress is also associated with reductions in biomass and crop yield that result in agricultural damage costing millions of dollars [6].

Several strategies have been developed to alleviate the effects of salt on plant growth and development, including plant genetic engineering [3] and, more recently, the use of phenolic compounds to promote growth [7]. Although the roles of phenolic compounds in plant growth promotion and nutrient management are not yet well known or established, there is a slowly emerging trend of using these compounds to overcome salt stress [8,9,10,11]. *p*-Coumaric acid (*p*-CA) is one of the phenolic compounds that have received much attention in the literature, with some studies suggesting that the utilization of *p*-CA is a promising alternative for alleviating salt stress [10,11]. A decrease in cell membrane permeability and reactive oxygen species levels alongside an increase in antioxidant enzyme and osmoprotectant levels were proposed by the cited authors to explain the positive effect of *p*-CA on increasing plant growth under salt stress.

However, apart from these studies, which have reported on physiological and biochemical changes, there seems to be a dearth of data on molecular or ion-omics studies. This lack of data is surprising when considering that plant ion-omics have provided insights into how plants respond to environmental stimuli, such as salinity, drought, or nutrient deficiencies, and how they regulate ion homeostasis to maintain optimal growth and development [12]. Thus, investigations into ion or mineral uptake involving phenolic acids holds promise for understanding how these compounds confer tolerance against salt stress. Remarkably, ion-omics studies on salt stress often show a strong correlation between salt stress tolerance and mineral uptake. Hence, in the current research, our aim was to study the influence of *p*-CA on the macro elements present in chia seedlings exposed to salt stress.

## 2. Results

### 2.1. p-CA Improves Chia Seedling Growth under Salt Stress

The results show that *p*-CA and salt stress (induced by the application of 100 mM of NaCl) differentially alter the shoot growth of chia seedlings (Figure 1). The exogenous application of *p*-CA enhanced shoot length by 26% compared to the control (Figure 1A). Salt stress negatively influenced shoot growth, as seen with the reduction in shoot growth of 14% relative to the control (Figure 1B). *p*-CA induced a reversal of the negative effects caused by salt stress. Shoot fresh weight was increased by 59% in response to *p*-CA, whereas a significant reduction amounting to 19% was observed in the salt stress treatment when compared to the control (Figure 1B). However, the salt-stressed plants supplemented with *p*-CA showed a marked increase in shoot length relative to the salt treatment and control plants (Figure 1B). A similar trend was observed for the shoot dry weights (Figure 1C). Exogenous *p*-CA enhanced shoot dry weight by 28% compared to the control, whereas a significant reduction of 17% was observed in the salt stress treatment. However, the application of *p*-CA to salt-stressed plants reversed the reduction in dry weights (to the level of the control) that resulted from the salt stress treatment.

### 2.2. Effects of p-CA and Salt Stress on Mineral Content

Na, K, P, Mg, and Ca content was measured in the shoots of chia seedlings in response to *p*-CA, salt stress, and a combination of *p*-CA and salt stress (Table 1).

#### 2.2.1. Na Content

The shoot Na content of the chia seedlings was significantly enhanced in all the treatments, with the highest increase observed in the combined treatment (*p*-CA + NaCl) when compared to the control. *p*-CA increased Na content by 322%, whereas salt stress increased Na content by 512%. However, the highest increase in Na content (1175%) was observed in the salt-stressed plants supplemented with *p*-CA. The increases in Na content in all the treatments were quantified relative to the untreated control (Table 1). We also observed a 108% increase in Na content in the Salt + *p*-CA treatment when compared to the Salt-only treatment (Table 1).

#### 2.2.2. K Content

The results showed that *p*-CA did not alter K content when compared to the control. However, a significant reduction in K content was observed in response to salt stress (NaCl). Salt stress reduced K content in the shoots of chia seedlings by 66% when compared to the control. The decrease in K content in the combined treatment was not as pronounced as that observed in the salt stress treatment. Salt-stressed chia seedlings supplemented with *p*-CA had 11% lower K content relative to the control (Table 1). However, an increase in K content of 158% was observed when comparing the Salt + *p*-CA treatment to the Salt-only treatment (Table 1).

#### 2.2.3. P Content

The P content in the shoots of chia seedlings was differentially altered by the different treatments. A slight but significant increase (14%) in P content was observed in response to *p*-CA. However, the chia seedlings subjected to salt stress and a combination of salt stress and *p*-CA showed a significant reduction in P content, with the more significant reduction observed in the plants subjected to the salt stress treatment (75%). In the combined treatment (*p*-CA + NaCl), P content was reduced by 35% relative to the untreated control (Table 1). Furthermore, an increase in P content of 164% was observed in the plants subjected to the Salt + *p*-CA combination treatment when compared to the increase induced by the Salt treatment (Table 1).

#### 2.2.4. Mg Content

Similar to what was observed for Na and P, exogenous *p*-CA increased Mg content to a level significantly higher (13%) than that observed for the untreated control. In response to salt stress, Mg content was reduced by 57% when compared to the untreated control. Contrary to what was observed in the salt stress treatment, a significant increase (22%) in Mg content was observed in the plants subjected to salt stress treatment supplemented with exogenous *p*-CA. This increase in Mg content was significantly higher than that observed for both the plants subjected to the *p*-CA treatment and the untreated control (Table 1). But we also observed a significant increase in Mg content amounting to 180% in the Salt + *p*-CA-combination-treated seedlings when compared to the Salt treatment group (Table 1).

#### 2.2.5. Ca Content

The Ca content in the shoots of chia seedlings was differentially altered by the various treatments. A significant increase (51%) in Ca content was observed in response to *p*-CA. However, the chia seedlings subjected to salt stress showed a significant reduction amounting to about 50%, while the combination of salt stress and *p*-CA resulted in a significant increase in Ca content, which was more pronounced, constituting a 56% increase compared to the control. When determining the role of *p*-CA in salt stress, there was an increase in Ca content of about 209% in the Salt + *p*-CA treatment compared to the plants subjected to salt stress without *p*-CA. This result shows that exogenous *p*-CA application increased all the microelement concentrations when comparing salt stress with or without *p*-CA, as indicated by the green arrows showing the increases due to the presence of exogenously applied *p*-CA in salt stress conditions.

## 3. Discussion

The obtained results illustrate that salt stress reduced the growth characteristics of chia seedlings when compared with those of the control plants. Our results are in harmony with those obtained by Jones et al. [9]. However, exogenously applied *p*-CA effectively resulted in enhancing plant growth characteristics (Figure 1). In this experiment, shoot length and fresh and dry weights were significantly reduced due to salt stress, while the exogenous application of *p*-CA significantly improved these parameters (Figure 1). These results may be attributed to *p*-CA’s role in improving plant biomass and chlorophyll content in Chia seedlings [7]. The simultaneous application of exogenous *p*-CA and salt stress was shown to reverse the effects of salt stress. A similar phenomenon was also observed in a study by Kaur et al. [11], which demonstrated that an exogenous application of *p*-CA reversed the effects of salt stress, as exhibited by the improvement in plant biomass growth. Besides the physiological effects, these studies focused on the biochemical changes (antioxidant enzyme and ROS metabolism) associated with *p*-CA-induced salt stress [10,11]. These results all seem to indicate that *p*-CA has a robust ability to minimize the effects of salt stress. However, the mechanism of action of *p*-CA-induced salt stress tolerance remains elusive. This brief report sheds further light on the possible involvement of essential macro elements when determining the role of *p*-CA as a growth-promoting signaling molecule, as we previously suggested [7]. This report further suggests that *p*-CA not only promotes growth under normal conditions but also reverses the effects of NaCI treatment.

A greater effect of salt stress on plants is known to be associated with higher increases in the content of sodium (Na) [13,14]. An excessive accumulation of intracellular Na leads to ion imbalance and toxicity in plants [15,16,17], and a positive balance in ions’ response to salt stress is required, so maintaining a positive ion balance in response to salt stress is essential. As expected, under salt stress, there was an increase in Na content, which ultimately reduced the content of all the other essential macro elements that were tested in the study (Table 1). Previous studies have also reported a phenomenon similar to that observed in our study (Table 1), demonstrating that under salt stress, there was a reduction in the levels of essential macro-elements such as K [18,19], P [20,21,22], Mg [21,23], and Ca [22,24]. However, while our previous results demonstrated that the inhibition of endogenous *p*-CA altered the reduction in the content of all the essential macro elements, which had a major impact on growth [25], in this study, we observed an opposite phenomenon when treating chia seedlings with exogenous *p*-CA. Part of this could further explain why we observed an increase in plant growth via the *p*-CA treatment [7]. After treatment with salt plus *p*-CA, we observed an increase in Mg and Ca levels. Concurrently, there was a reversal in the effect of salt stress on plant biomass (Figure 1). A previous study by Tester and Davenport [26] reported that a reduction in shoot growth under salt stress could be attributed to the excessive accumulation of Na; although we observed the highest increase in Na in our NaCI + *p*-CA treatment, we also observed and increase in growth, suggesting that Mg and Ca might play an essential role in conferring tolerance.

In conclusion, this work further illustrates the complexity of the mechanisms underlying *p*-CA’s role in growth and development, as, in our previous study, we demonstrated that amongst the several possible pathways through which *p*-CA controls growth was via the activation of an ROS-signaling pathway involving O_2_^∙−^ under the control of proline accumulation [7]. Khairy and Roh [10] also studied the mechanism through which *p*-CA in conjunction with salt stress concentrations effects growth and non-enzymatic (proline and vitamin C) antioxidant capacity. They concluded that *p*-CA in combination with salt stress did not show significant differences compared to the control. However, their study had a critical design flaw, as they failed to include a treatment group subjected to salt stress only, which would have helped in demonstrating if *p*-CA was able to reverse the effect of salinity stress and to what level. Kaur et al. [11] investigated the mitigation of salinity-induced oxidative damage through the application of phenolic acids. Their findings showed that *p*-CA effectively reduced salt stress and enhanced wheat seedling growth through various mechanisms. Specifically, the model of action proposed was that *p*-CA decreases the accumulation of reactive oxygen species (ROS) and malondialdehyde (MDA) by controlling antioxidant capacity. Alternatively, it was also suggested that *p*-CA may also regulate Na content and alleviate salt stress damage by reducing ion leakage [11,27]. This discovery highlights the importance of antioxidant homeostasis and suggests the involvement of ion-omics in the tolerance mechanism, which is mediated by *p*-CA’s control of ion leakage.

Our current study builds upon this research by exploring additional mechanisms involving ion-omics in chia seedlings, subjects that were not previously explored. Based on physiological and ion-omics data, we found that there was a strong correlation between the elements Mg and Ca and plant growth. This may suggest that chia seedling growth and adaptation to salt stress may occur through the ionic homeostasis of Mg and Ca rather than Na. This is because, though we observed higher levels of Na in the combined (NaCI + *p*-CA) treatment than the salt stress treatment, the combined treatment seemed to reverse the physiological growth effect caused by salt stress (Figure 1; Table 1). Hence, we suggested that both Mg and Ca may be the major essential macro elements contributing to the growth of chia seedlings, as their levels significantly increased in the combined treatment. Thus, from the above discussion, it can be gleaned that *p*-CA plays a very critical role in plant growth and the mitigation of salt stress in chia seedlings. The findings indicate that *p*-CA can significantly enhance Ca and Mg content and support plant health under salt stress conditions, possibly because Ca and Mg are vital in regulating the metabolic pathways and enzyme activities of amino acid synthesis. While this study focused on the seedling shoots, future research could extend to non-soil experiments to analyze root responses more accurately. Making sequencing data of chia genomes available would offer insights into the genetic basis of *p*-CA responses, allowing us to identify specific genes and regulatory elements involved in this pathway. This knowledge could reveal how *p*-CA modulates gene expression to counteract the adverse effects of salt stress. Additionally, utilizing further omics tools such as metabolomics and proteomics could identify crucial molecular, metabolite, and protein targets of *p*-CA signaling pathway. This could elucidate the biochemical and cellular processes through which *p*-CA exerts its protective effect against salt stress.

## 4. Materials and Methods

### 4.1. Plant Growth and Treatment

This study was conducted at the University of the Western Cape in a greenhouse during the 2019 season. Geographically, the experimental site is located at 33°55′51.24″ S latitude and 18°37′28.54″ E longitude at an altitude of 131.2 feet above sea level. The area falls under the Swartland agroecological region, characterized by its Mediterranean climate, with the highest recorded winter rainfall amount being 96 mm (June) and the lower summer rainfall amount being 23 mm (February), experiencing an annual rainfall amount of 450 mm.

The Chia seeds used for the greenhouse experiment were purchased from Faithful to Nature, Cape Town, South Africa, in the 2019 season. A total of 400 seeds were soaked on wet filter paper to allow them to germinate in the dark for a period of 72 h [7]. Germinated seeds were transferred to 19/20 cm plastic pots filled with moist promix growth medium (Stodels Garden Centre, Brackenfell, South Africa) and allowed to develop into sprouts. Chia sprouts of similar height were selected for all experiments. Control plants were supplemented with 50 mL of Nitrosol^®^ (Auckland, New Zealand) solution (a natural organic liquid foliar feed formulation for use on plants that is not harmful to bees, birds, or other animals when used directly) diluted in water (1:300) containing 0.001 M of NaOH. Three sets of experiments containing nitrosol^®^ Solution were prepared: *p*-CA (100 µM of *p*-coumaric acid), salt stress (100 mM NaCl), and a combination of 100 mM of NaCl and 100 µM of *p*-CA). The experiment was designed using a factorial layout, with a complete randomized design to mitigate the effect of environmental variations across different positions in the growth room. Plants were treated until the initial stages of V_2_ of vegetative growth under a 27/19 °C day/night temperature cycle under a 16/8 h dark cycle at a photosynthetic photon flux density of 300 μmol photons m^−2^·s^−1^ during the day phase. Seedlings’ shoots were harvested and then measured for fresh and dry weights. Meanwhile, remaining seedling shoots were harvested by snap-freezing the shoots with liquid nitrogen and then grinding the seedling shoots for long-term storage.

### 4.2. Measurement of Plant Growth

Chia seedling growth was measured using a method that was first described by Gokul et al. [28] and modified by Nkomo et al. [7]. Chia seedlings were carefully extracted from the promix growth medium to avoid damaging the shoots and roots when uprooting the chia seedlings. The roots were separated from the shoots to prevent erroneous data interpretation caused by possible root damage. The chia seedlings from each treatment were scored with respect to shoot length (SL), fresh weight (FW), and dry weight (DW) of shoots. The DW was determined by drying the seedlings in an oven at 55 °C for 48 h, as described by Gokul et al. [28].

### 4.3. Measurement of Inductively Coupled Plasma Optical Emission Spectroscopy Analysis

The shoots’ multi-element compositions were analyzed using inductively coupled plasma optical emission spectroscopy (ICP-OES), referred to as the “ionomics” approach [12]. Frozen plant material (150 mg) from Section 4.1 was digested according to the method reported by Huang et al. [29], and the frozen plant material was inserted into 2 mL Eppendorf tubes, to which 1 mL of 65% nitric acid was added. The resulting solution was then mixed through vigorous shaking via a vortex machine and then incubated at 90 °C on a heating block for 4 h to allow sample digestion. After they were digested, the samples were centrifuged, forming a pellet consisting of all residual material, and the supernatant was transferred to a new tube. A 10X dilution was conducted in a 5 mL final volume using 2% nitric acid as a diluent. The product was then analyzed an ICP-OES machine. The digested chia seedling shoot material was processed according to Vachirapatama and Ji-rakiattikul [30]. The concentrations of four essential macro elements (Na, P, K, and Mg) were determined using a Varian Vista Pro CCD simultaneous inductively coupled plasma optical emission spectrometer (Varian, Sydney, Australia) with certified standards (Sigma, St. Louis, MO, USA; TraceCERT^®^).

### 4.4. Statistical Analysis

All experiments described were performed six times independently. For measurement of plant growth parameters (shoot fresh and dry weights), 30 individual chia seedlings (grouped in pools of 10) per treatment were analyzed. For all other experiments, 50 chia seedling shoots (grouped in pools of 10) were homogenized per treatment. For statistical analysis, the one-way analysis of variance test was used for all data, and the means (for six independent experiments) were compared according to the Tukey–Kramer test at 5% level of significance, using GraphPad Prism 5.03 software.

## Figures and Tables

**Figure 1 plants-13-01564-f001:**
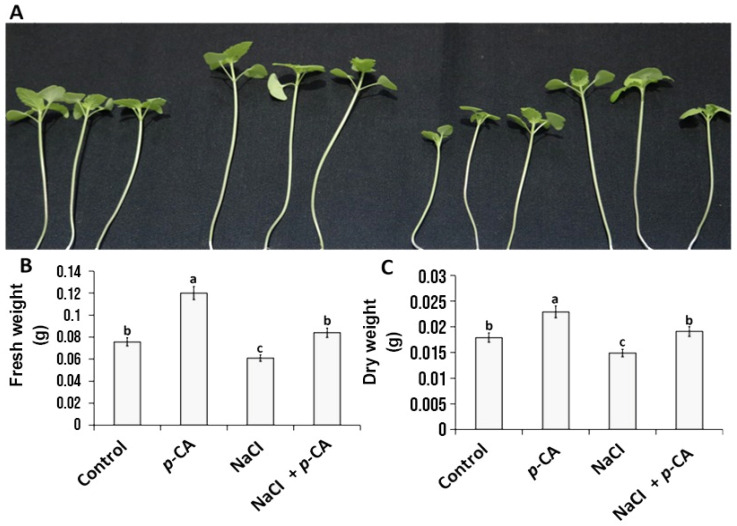
The influence of *p*-CA on chia seedling biomass under salt stress. Plant growth parameters include triplicate representatives of each treatment (**A**) and shoot (**B**) fresh weights as well as shoot (**C**) dry weights. Data represent the means (±SE) from six independent experiments. Different letters represent statistical significance at *p* < 0.05 (with respect to the Tukey–Kramer test).

**Table 1 plants-13-01564-t001:** Interactive influence of *p*-CA and salt stress on essential macro elements in the shoots of chia seedlings. Macro element data expressed in mg·g^−1^ FW, expressed as means ± SE. The blue arrow represents an increase in macro element content, while the red arrow represents a decrease, and the NS sign shows that no significant difference was observed when compared to the control. The green arrow represents an increase in macro element content when comparing Salt + *p*-CA to salt stress only.

Minerals	Mineral Relative Content (mg·g^−1^ FW)			
	Control	*p*-CA	Salt	Salt + *p*-CA
Na	0.057 ± 0.004	0.241 ± 0.194 ↑	0.349 ± 0.015 ↑	0.727 ± 0.067 ↑↑
K	4.513 ± 0.068	4.326 ± 0.270 ^NS^	1.552 ± 0.062 ↓	4.007 ± 0.707 ↓↑
P	0.329± 0.002	0.376 ± 0.064 ↑	0.081 ± 0.004 ↓	0.214± 0.012 ↓↑
Mg	0.387 ± 0.006	0.439 ± 0.035 ↑	0.168 ± 0.007 ↓	0.471 ± 0.031 ↑↑
Ca	0.497 ± 0.002	0.751 ± 0.200 ↑	0.250 ± 0.008 ↓	0.773 ± 0.039 ↑↑

## Data Availability

The authors confirm that the data that support the findings of this study are available within the article.

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
