# Peer review of "p-Coumaric Acid Differential Alters the Ion-Omics Profile of Chia Shoots under Salt Stress"

_plants, 2024, doi:10.3390/plants13111564_

Round 1
Reviewer 1 Report
Comments and Suggestions for Authors
Materials and methods section needs to be expanded considerably. How many plants, when, where, how, etc. How big were plants at harvest? What is ionomics? Describe the treatments more clearly. What is Nitrosol? What is p-coumaric acid? Why are you using it? What does it do to salt in plants?
Comments on the Quality of English LanguageYou use "while" to open sentences. That makes the following words a phrase, not a sentence. You do this in several places in the text.
L. 96-98: Should this read "P"? You have K.
L. 111: an asterisk always indicates significance in scientific writing. NS indicates not significant.
L. 182-187: Add more information on plant size
Reviewer 2 Report
Comments and Suggestions for Authors
The experimental design of this article is too simplistic, and the chart display in the article does not comply with academic standards. It is recommended to reject the manuscript.
Comments on the Quality of English Languagenot avaiable
Reviewer 3 Report
Comments and Suggestions for Authors
In this Short Communication, the application of phenolic compounds to salt tolerance was studied, and the results were clear.However, there are some areas that can be improved. For example, firstly, there is too little explanation on the application of phenolic substances in the introduction.Secondly, there is a lack of discussion on the internal mechanism of existing results.In addition, for the data description is slightly less scientific, it is suggested to use significance to express the data, for example, the data differences in the table can be expressed by the results of multiple comparisons instead of arrows.
Round 2
Reviewer 1 Report
Comments and Suggestions for Authors
Paper is difficult to read. English needs to be edited for clarity.
L. 36-37. rewrite eg: results in agricultural losses of millions of dollars
L. 59-74. rewrite
L. 335 ff. Statistics. Explain how the six experiments were combined for the statistical analysis.
Comments on the Quality of English LanguageThe paper needs to be edited by a native English speaker or AI program. There are too many grammatical errors for a reviewer to correct them.
Author Response
We have carefully review and revise the manuscript and also ask the native English speaker in biotechnology department at University of the Western Cape to ensure that our writing style is more precise and straightforward, adhering to the formal tone required for scholarly work. We have reworked the statistical analysis for clarity. Thank you again for your constructive comments.
Reviewer 3 Report
Comments and Suggestions for Authors
Can be accepted.
Author Response
We Thank the reviewer for recognising the importance of our work and suggesting it for publication.